# Orientation-Encoding CNN for Point Cloud Classification and Segmentation

**Hongbin Lin [1],\*, Wu Zheng [2] and Xiuping Peng [2]**

[1]  School of Electrical Engineering, Yanshan University, Qinhuangdao 066004, China
[2]  School of Information Science and Engineering, Yanshan University, Qinhuangdao 066004, China;
    zweducn@163.com (W.Z.); pengxp@ysu.edu.cn (X.P.)
\*  Correspondence: honphin@ysu.edu.cn

**Abstract:** With the introduction of effective and general deep learning network frameworks, deep learning based methods have achieved remarkable success in various visual tasks. However, there are still tough challenges in applying them to convolutional neural networks due to the lack of a potential rule structure of point clouds. Therefore, by taking the original point clouds as the input data, this paper proposes an orientation-encoding (OE) convolutional module and designs a convolutional neural network for effectively extracting local geometric features of point sets. By searching for the same number of points in 8 directions and arranging them in order in 8 directions, the OE convolution is then carried out according to the number of points in the direction, which realizes the effective feature learning of the local structure of the point sets. Further experiments on diverse datasets show that the proposed method has competitive performance on classification and segmentation tasks of point sets.

**Keywords:** point clouds; orientation-encoding (OE) convolution; local geometric feature; classification; segmentation





## 1. Introduction

At present, deep learning has achieved significantly success in image recognition tasks, such as image classification [1–3] and semantic segmentation [4,5]. The rapid development of two-dimensional data fields has promoted researchers' interest in three-dimensional data recognition and segmentation tasks. With the extensive application of 3D laser scanners and 3D depth sensors, algorithms for the effective analysis of point cloud data are required in terms of autonomous driving, robots, unmanned aerial vehicles, and virtual reality. It is not always feasible to directly apply the two-dimensional image deep learning methods to the three-dimensional data tasks, because in the three-dimensional scene composed of the point cloud, these point set objects are disordered and scattered in the three-dimensional space. It is also unreasonable to simply apply two-dimensional features to irregular point clouds through convolution operators, because these operations are carried out on regular grids. The methods of [6–8] try to address this problem by using three-dimensional convolutional neural network voxelization scenarios. However, as the main challenges of voxel representation are spatial sparsity and computational complexity, the researchers in [9,10] try to use special methods (such as octree) to solve the sparsity problem, but it takes a certain amount of time to convert the point cloud into voxels.

Because of the limitations of the various above explorations, the PointNet [11] structure directly uses the point cloud as the input data and then uses the T-net module to convert the input point cloud to solve the problem of rotation invariance of the point cloud object, combined with a Multi-Layer Perceptron (MLP) to extract the high-level semantic information of the input data object, and finally max pooling to extract the global information. The PointNet architecture solves the problem of point cloud disorder and produces a general network architecture for directly processing the point cloud data. However, the

local geometric features of point cloud objects are not taken into account in the network architecture when extracting high-level semantic information. Afterwards, PointNet++ [12] downsamples the sample data by means of the Farthest Point Sampling (FPS) algorithm and uses the ball query algorithm to search the samples for a set of adjacent points within a certain range; then the original features of the combined point sets learn high-level semantic features through convolution operations. The core idea is to propose a hierarchical structure, which solves the defects of PointNet local feature extraction and further improves the performance of the network.

PointNet [11] and PointNet++ [12] are the first deep network frameworks for point set processing, and several studies have promoted this research direction by proposing improvements in structure or composition [13,14]. Considering the relative layout of adjacent points and their features, a new pooling strategy is combined to carry out spectral convolution on local graphs [13]. SpiderCNN [14] proposes a convolution kernel with parameterization by learning the weighting parameters from the features of the input point sets. These methods attempt to enrich feature sets through original point cloud data features to enhance the performance of point cloud classification and segmentation tasks. However, these schemes still have problems such as insufficient extraction of local features of the point cloud and poor universality and robustness of the network architecture; hence, the 3D point cloud data task is still a long-term and challenging process.

In this paper, we propose a new orientation-encoding convolutional neural network (OECNN) for the point cloud data. In order to overcome the problems of low accuracy and poor robustness of the network architecture, we adopt a special convolution method and a pooling strategy. Our main contributions in this paper are as follows:

- We propose a general network architecture for point cloud classification and segmentation.
- The framework is simple and effective.
- The network has certain adaptability.
- Our OE convolution and pooling strategies are perceptive to local geometric features of point sets.

## 2. Related Work

### 2.1. Point Cloud Classification and Segmentation

In the point cloud model, each sample is composed of point sets. Point cloud classification can be stated as follows: Given a set of sample points in three-dimensional space, we learn the high-level semantic feature information of samples through neural network to match the sample label. Each sample matches the corresponding label, which is an end-to-end supervised learning process. The point cloud segmentation task is a further extension of the classification tasks, and its purpose is to match the category label of each point in the sample. As we have entered the era of big data, deep learning has been widely studied through the application of optimization algorithms in neural networks and various tasks that take point clouds as the research object have attracted researchers' attention.

### 2.2. Voxel Data

Voxel data is a regular data structure which is easy to process. VoxNet [7] and NormalNet [15] apply 3D convolution to a voxelization of point clouds. However, there are high computational and memory costs associated with using 3D convolution. A variety of work [9,16] is devoted to exploring the sparsity of voxelized point clouds to improve the efficiency of computing and memory. OctNet [9] uses the sparsity of the input data to divide the space using a series of unbalanced octrees, and each leaf node in the octree stores a pooled feature representation. This representation focuses on memory allocation and computation in the relevant dense regions and enables deeper networks to handle higher resolutions. The Sparse Submanifold CNN architecture [16] proposes sparse convolution operations to deal with spatial sparse data more effectively and use them to construct

spatial sparse convolutional networks. In comparison, our OECNN is able to directly use point clouds as input data and process very sparse data.

*2.3. Spatial Domain*

The GeodesicCNN [17] is a generalization of the convolution network paradigm to non-Euclidean manifolds. Its construction is based on a local geodesic system consisting of polar coordinates to extract "patches", and the coefficients of the filters and linear combination weights are optimization variables that are used to learn to minimize specific cost functions.

An image is a function on regular grids $F : R^2$. Let $W$ be a $(2m + 1) \times (2m + 1)$ filter matrix, where $m$ is a positive integer. The convolution in classic CNNs is:

$$F * W(i, j) = \sum_{s=-m}^{m} \sum_{t=-m}^{m} F(i - s, j - t)W(s, t) \tag{1}$$

GeodesicCNN uses the patch operator D to map a point $p$ and its neighbors, and then applies Equation (1). The method learns the influence of patch operation in the local polar coordinate system of point $p$. We offer an alternative viewpoint; instead of finding local parametrizations of the manifold, we view it as an embedded Euclidean space in $R^n$ and design convolution methods. Our method is more efficient for point cloud processing in the Euclidean space.

## 3. Method Design

We studied a series of different convolution operations [13,14,18] and pooling methods [14] for point cloud data. In PointSIFT [18], the authors proposed an operator with orientation-encoding and scale perception. They search eight nearest points for each point in eight directions and extract the features of point sets through three-layer convolution (PointSIFT convolution will carry out three-layer convolution according to three directions, xyz) and max pooling. However, when searching each nearest point, all the input points need to be traversed. The system also has some limitations in that only the eight nearest points can be searched for each point.

Unlike PointSIFT, in this paper, we proposed an orientation-encoding operator and carry out effective convolution in each direction. We divide the spherical region in a certain range into eight directions, search the same number of points in each direction for each point, and sort the searched point sets according to the direction. After that, we extract the corresponding features of point sets through a two-layer convolution operation (the OE convolution is convolved by the number of points in each direction) and top-k pooling strategy. The method in this paper can adjust the scale (such as radius and the number of points) according to the features learned from convolutional blocks by orientation-encoding, which has certain adaptability. Moreover, convincing experimental results have been obtained on ModelNet40 and ShapeNet part datasets. PointSIFT convolution and OE convolution are shown in Figure 1.

*3.1. Orientation-Encoding (OE) Architecture*

We present our OE convolution module in this section. In order to capture shape patterns adaptively, we hope that shape information can be clearly encoded in different directions. Hence, we propose a new orientation-encoding convolution for all point operations. As illustrated in Figure 2a, PointSIFT can ideally search for the 8 nearest points (red points) in 8 directions for each point in the cube. However, this search method has a large fault tolerance and increases the computational complexity. Our OE search can selectively search for the desired number of points in 8 directions of the spherical area, with some flexibility, and better represents the surrounding point set features. Figure 2b shows searching for 4 points in each of the 8 directions.

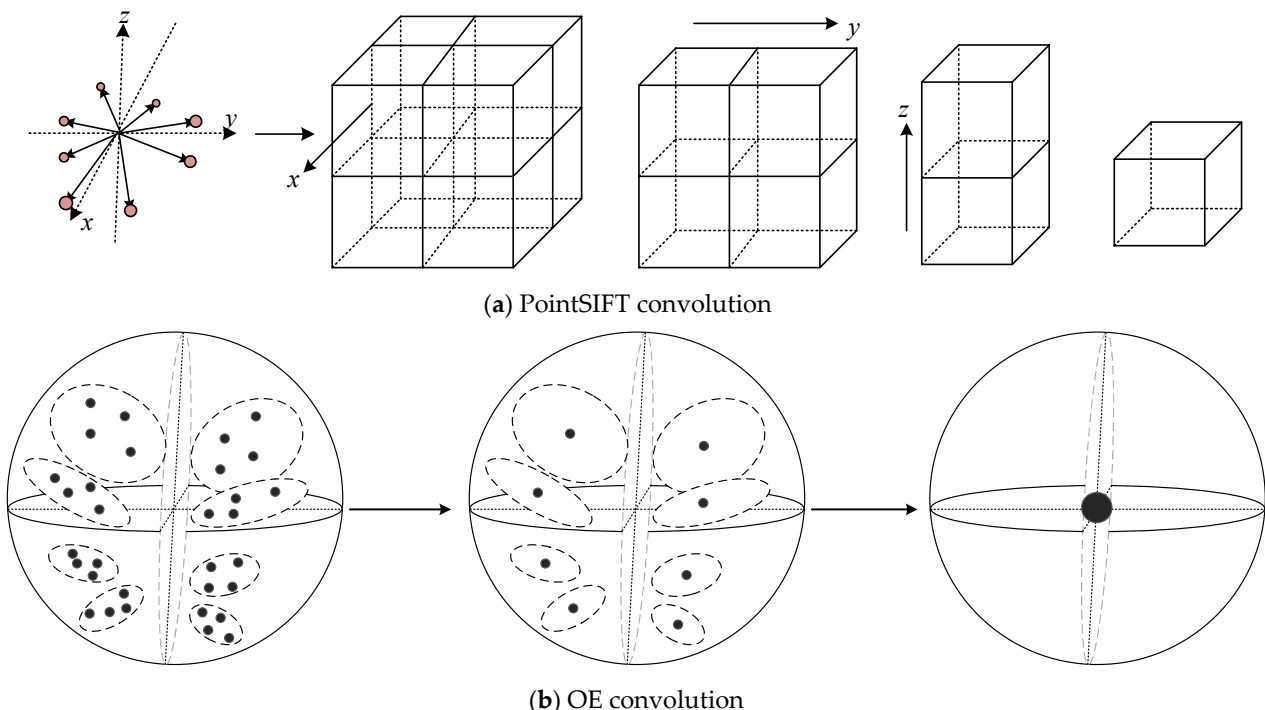

(**a**) PointSIFT convolution

(**b**) OE convolution

**Figure 1.** The comparison of PointSIFT and OE convolution.

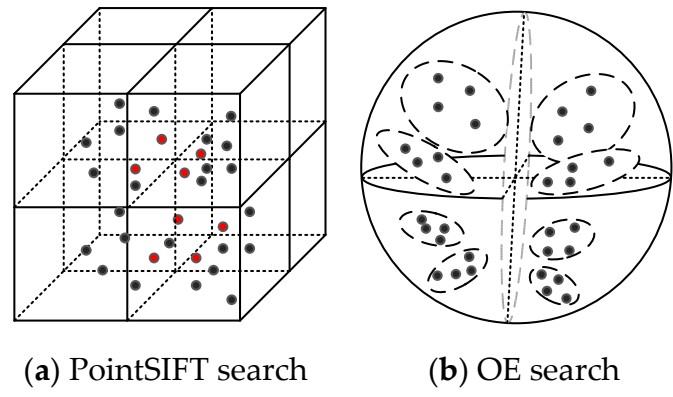

(**a**) PointSIFT search          (**b**) OE search

**Figure 2.** PointSIFT and OE search point methods.

In three-dimensional space, with an input of $n$ points with $d$ dimension features, for each point $p_0$ (with $d$ dimensions), the 3D space is divided into 8 partitions with $p_0$ as the center, indicating 8 directions. We define variables for each direction to store the index of the points to be searched in each direction and define the corresponding indicators to indicate the number of local points to be searched in each direction. In the spherical region with radius $r$, we find $m$ points for $p_0$ in each direction (let the number of local points searched in the space of each point be M, $m = M/8$). $m$ points represent local geometric features in one direction.

Before the training, we rotate, jitter, and randomly select a fixed number of the sample data in each epoch; hence, the search of local points around the central point $p_0$ can be regarded as a random process. When we have searched the corresponding number of points within the radius $r$, then we do not need to search other points. In the process of searching local points for each central point, the worst situation is to traverse all sample points once. In PointSIFT, because the search objective is to find the nearest point in each direction, all points in the sample need to be traversed eight times. Therefore, in theory, our method has a certain speed advantage and reduces the computational complexity. We can adjust the search range according to the radius $r$, so as to better capture the local

information in each direction. In order to prevent not searching enough points in radius $r$, we use the point $p_0$ to initialize the required number of points (with $d$ dimensions).

As shown in Figure 3, we propose the OE convolution module, which has two paths to extract local high-dimensional semantic information for each point in the sample. On the one hand, we first use OE search to find the local point sets and the corresponding features for each point and store the corresponding information by increasing the one dimension. In order to make the convolution orientation-aware, we conduct a two-layer convolution. The first layer of convolution is performed according to the number of points in each direction to obtain the remaining 8 points in 8 directions, with one point in each direction. The second convolution convolves the remaining 8 points, and then reduces the dimensions to obtain the corresponding point set features. We used the same output channel $e$ for two convolutions. At this point, each point has local high-dimensional semantic feature information. On the other hand, we use the input point set features to directly perform the convolution with the output channel $e$ to obtain the high-dimensional semantic features of each point. After that, we obtain richer local point set high-dimensional semantic information by performing an addition operation.

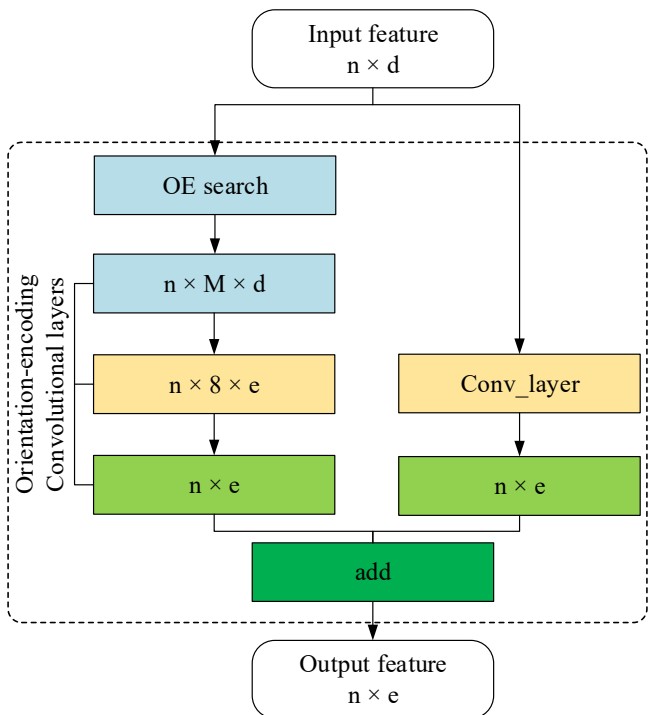

**Figure 3.** OE convolution module.

We put the obtained feature of points into a tensor $S \in R^{n \times M \times d}$. The two stages of directional convolution are:

$$S_1 = g[Conv_m(A_m, S)] \in R^{n \times m \times e}$$
$$S_2 = g[Conv_8(A_8, S)] \in R^{n \times 1 \times e} \qquad (2)$$

where $A_m$ and $A_8$ represent the weight parameters to be optimized, $Conv_m$ represents the convolution of $m$ points along each direction, $Conv_8$ represents the convolution of the remaining eight points in eight directions. In this paper, we set $g[\cdot] = ReLU[Batch\_norm(\cdot)]$. After the convolution, each point is represented as a vector with $e$ dimensions. This vector represents the shape pattern around $p_0$.

### 3.2. Multi-Scale Architecture

Using an OE convolution module as a basic unit, we are able to build a multi-scale network structure. An OE convolution module can capture arbitrary scale information from eight directions and select any number of points in each direction. If we stack several OE convolution modules to generate a deeper network structure, then the last layer can observe a larger three-dimensional region, and different OE units can have different scales. As illustrated in Figure 4, we can choose the appropriate scale and the number of points according to the features of the network and strive to better optimize the performance of the network. A simple but effective way to capture multi-scale patterns is to concatenate the output of different stacked units as a shortcut.

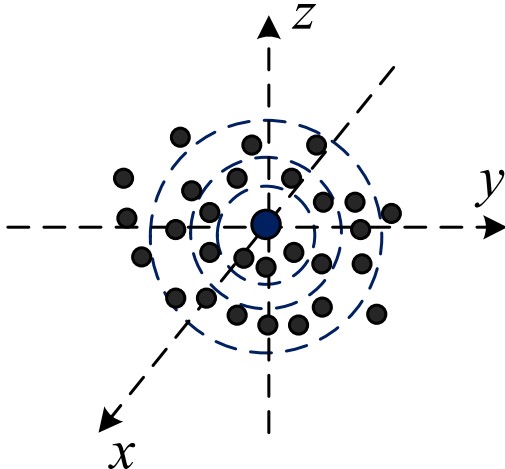

**Figure 4.** Multi-scale selection.

For a layer of OE convolution module, searching for M local points for each point in the sample can be regarded as a random process. When we stack OE convolution modules, if we use the same scale and search the same number of local points for each layer of OE convolution module but with different numbers of output channels, the M points searched by each layer of OE convolution module are the same. In this way, we can use neural networks to learn different high-dimensional semantic information of the same local point set in a certain range, and finally fuse the feature information by concatenation. If we use different scales and different numbers of local search points in each layer of the OE convolution module, then the M points in each layer of the OE convolution module are different, which will lead to a different local scope and different local semantic feature information collected by each layer of OE convolution module, which will generate information redundancy. This is not conducive to the feature learning of the local range; hence, we set the same scale and the same number of local points in each layer of the OE convolution module to generate more representative local high-dimensional semantic information for experiments. In the following sections, we also conducted comparative experiments on multi-scale and fixed-scale structures.

### 3.3. Top-k Pooling vs. Max Pooling

Max pooling can be seen as a special type of top-k pooling. By applying max pooling, we can extract global point cloud features. However, because it does not have certain scalability and will lose data information, we adopt a selective top-k pooling strategy proposed in SpiderCNN [14]. Both max pooling and top-k pooling use a simple symmetric function to gather information from each point. Here, a symmetric function takes $n$ vectors as inputs and outputs a vector representing global point cloud information in a sample which is invariant to the input order.

Our idea is to generate a function that can extract global features by applying a symmetric function in the feature space of a point set:

$$f_{\max\ pooling}(\{x_1,\dots,x_{n-1}\}) \approx h_1(Conv(x_1),\dots Conv(x_{n-1})) \tag{3}$$
$$f_{top\_k\ pooling}(\{x_1,\dots,x_{n-1}\}) \approx h_2(Conv(x_1),\dots Conv(x_{n-1}))$$

where *h* is composed of a single variable function and max pooling (or a top-k pooling). *f* is the corresponding sample features, and the number of features of $f_{top\_k\ pooling}$ is *k* times that of features of $f_{\max pooling}$. The value of *k* determines that top-k pooling has good selectivity. Through the collection of *h*, we can learn a number of features to capture different properties of the set in different directions. Under the same experimental conditions, we compare the two pooling methods on ModelNet40 [19]. The max pooling classification accuracy is 92.2%, and the top-k pooling classification accuracy is 92.5% when the value of *k* is 2, which reflects the advantages of top-k pooling in extracting global feature information. In Figure 5, we use the 2 × 2 matrix to give the calculation process of max pooling and top-k pooling and show the selectivity of top-k pooling.

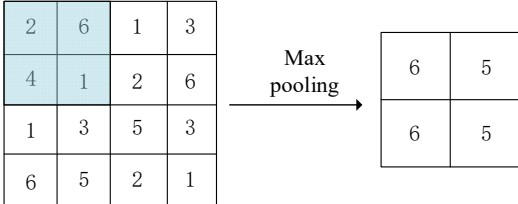

**(a)** Max pooling

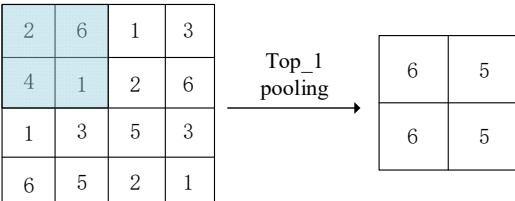

**(b)** Top_1 pooling

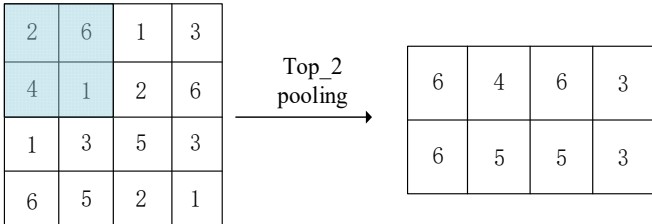

**(c)** Top_2 pooling

**Figure 5.** Calculation process of max pooling and top-k pooling.

## 4. Experiment

### 4.1. Experimental Environment

We evaluated and analyzed the OE convolution (OEConv) module on the 3D point clouds classification and segmentation. Through the 4-layer OE structure, we empirically studied the key parameters and compare our model with the state-of-the-art methods. All models were constructed with Tensorflow 1.5 on 1080Ti GPU and trained using the Adam optimizer with a learning rate of 0.001. The same data augmentation strategy as for PointNet [11] was applied: the point cloud was randomly rotated along the up-axis and the

position of each point was jittered by a Gaussian noise with zero mean and 0.02 standard deviation. The system used was Ubuntu 16.04. A dropout rate of 0.5 was used with the fully connected layer. Batch normalization was used at the end of each OE convolution module with the decay set to 0.5 or 0.7. On a GTX 1080Ti, the forward-time of a OEConv layer (batch size 16) with in-channel 32 and out-channel 64 was 0.052 s. For the 4-layer OECNN (batch size 16), the total forward-pass time was 0.615 s.

### 4.2. Classification on ModelNet40

ModelNet40 [19] contains 12,311 CAD models of 40 categories, sampled into point clouds. We used the official split, with 9843 training and 2468 testing examples. Experiments took the (x, y, z) coordinates and normal vectors of the 1024 points as the input for OECNN on ModelNet40. Figure 6 illustrates the OECNN with 4 layers of OEConv, and the number of output channels for each layer of convolution was 32, 64, 128, and 256, respectively. The ReLU activation function was used after convolution. The output features of the four OEConvs were concatenated in the end. Top-k pooling of all the points was used to extract global features. The experiments showed that a 1-layer OECNN with a OEConv of 32 channels, 0.2 scale, and 16 the number of search points can achieve a classification accuracy of 88.4%, and the performance of the OECNN improves with the increasing number of layers of OEConv.

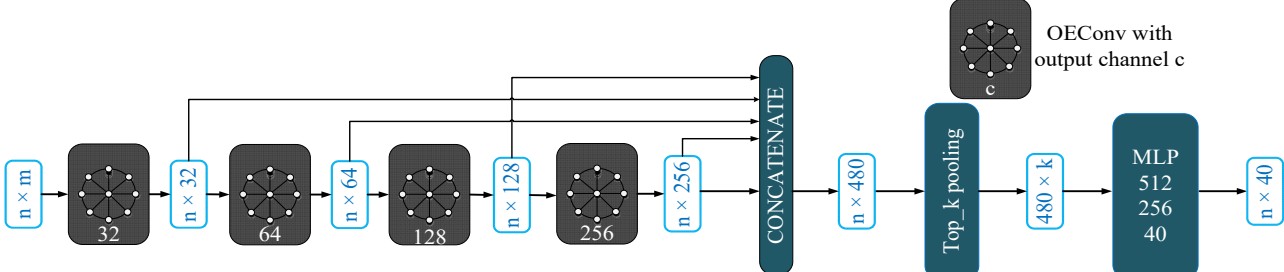

**Figure 6.** The architecture of a 4-layer OECNN in ModelNet40 [19] classification.

We compare three key parameters (the number of search points, different scales, and different values of k for top-k pooling) to improve the performance of the optimized network by using the single variable principle. The results are summarized in Figure 7. We saw that 16 is the optimal choice among 8, 16, 24, and 32 search points, and we chose a scale of 0.2 with top-2 pooling to get an accuracy of 92.5%. Then we used a fixed-parameter module to stack a 4-layer network structure, using top-4 pooling to get the best accuracy of 92.7%. We use a 4-layer multi-scale structure with different key parameters, and the classification accuracy is 92.6%. The result is slightly worse than for a 4-layer single-scale network with fixed parameters. We suspect that it may be due to the insufficient local features extracted from the multi-scale structure. To prevent overfitting, we apply the data augmentation method DP (random input dropout) introduced in [12] during training. Table 1 shows a comparison between OECNN and other models on ModelNet40. We also added the convolution operator proposed by PointSIFT into the OECNN network for comparison, and the result was only 90.3%. The 4-layer OECNN achieved an accuracy of 92.7%, which improves over the best reported result of models with 1024 input points. In Figure 8, we give a visualization of the misclassified samples of the two categories. We find that the reason for the misclassification is that they all have similar 3D geometric spatial features.

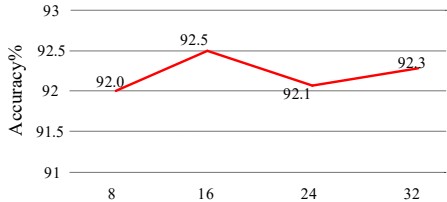

(**a**) The number of search points

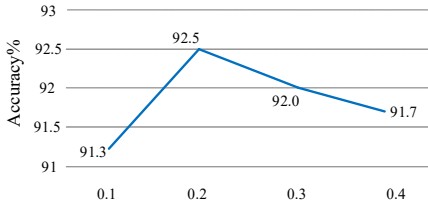

(**b**) Different scales

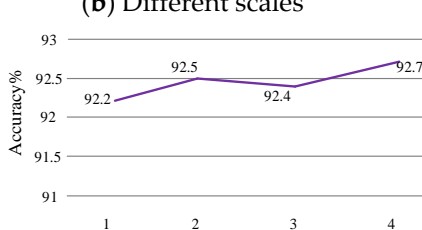

(**c**) Different values of k for top-k pooling

**Figure 7.** The comparison of key parameters in classification.

**Table 1.** Classification accuracy of OECNN and other models on ModelNet40 [19].

| Method | Points | Accuracy (%) |
|---|---|---|
| PointNet [11] | 1024 | 89.2 |
| PointNet++ [12] | 5000 | 91.9 |
| SpecGCN [13] | 2048 | 92.1 |
| SpiderCNN [14] | 1024 | 92.4 |
| PointSIFT+OECNN [18] | 1024 | 90.3 |
| DeepSets [20] | 5000 | 90.0 |
| Kd-Network [21] | 1024 | 90.6 |
| Pointwise CNN [22] | 1024 | 86.1 |
| PointGrid [23] | 1024 | 92.0 |
| PointCNN [24] | 1024 | 92.2 |
| DGCNN [25] | 1024 | 92.2 |
| OECNN | 1024 | **92.7** |

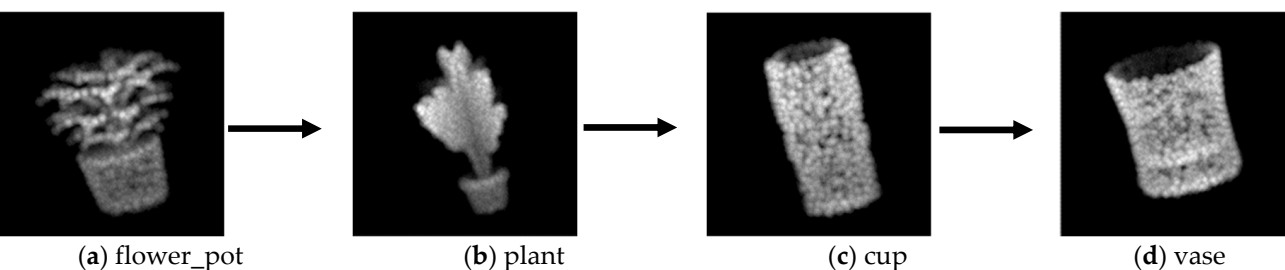

(**a**) flower_pot      (**b**) plant      (**c**) cup      (**d**) vase

**Figure 8.** The visualization of misclassified samples on ModelNet40 [19].

### 4.3. Segmentation on ShapeNet Parts

The ShapeNet Parts segmentation dataset [6] contains 16,881 shapes from 16 classes, with the points of each sample labeled into one of 50 part types. We used the official training/testing split with 14,006 for training and 2847 for testing. The challenge of the task is to assign a part label to each point in the test set. The mIoU (mean intersection over union) as the evaluation metric is the average of all part categories. As shown in Figure 9, like classification, we also compared three key parameters in the segmentation task. We used an OECNN with one layer of OEConv (the output channel is 64) to explore the learning situation of local features and compare the impact of different scales, the numbers of search points, and the value of k for top-k pooling. We found that the best result was 85.01% using a radius of 0.2, 24 search points, and top_2 pooling. Then we used a radius of 0.2, 24 search points, and top_2 pooling to stack an OEConv structure into a 4-layer OECNN structure. The structure shown in Figure 10 was trained with a batch size of 16. We used the point coordinates as the input and assumed that category labels were known. The experimental results are summarized in Table 2. We see that the OECNN network structure achieved competitive experimental results on the ShapeNet Parts dataset.

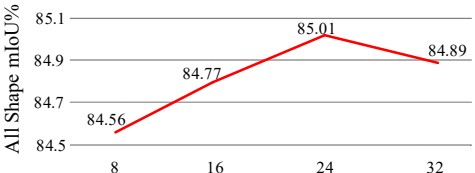

(**a**) The number of search points

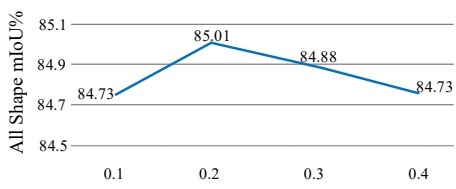

(**b**) Different scales

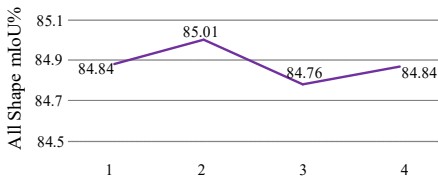

(**c**) Different values of k for top-k pooling

**Figure 9.** The comparison of key parameters in segmentation.

In comparison to the other five methods in Table 2, our method has a mIoU of 85.5% for all shapes on the ShapeNet dataset, and 10 categories of mIoU are superior to the other methods. Further, we tested an implementation of our model with the operator proposed for PointSIFT; its mIoU only reached 84.9%. Based on our analysis, we conclude that our method is sensitive to local information with a similar spherical shape because our OEConv module is able to capture local scale information in any spherical range.

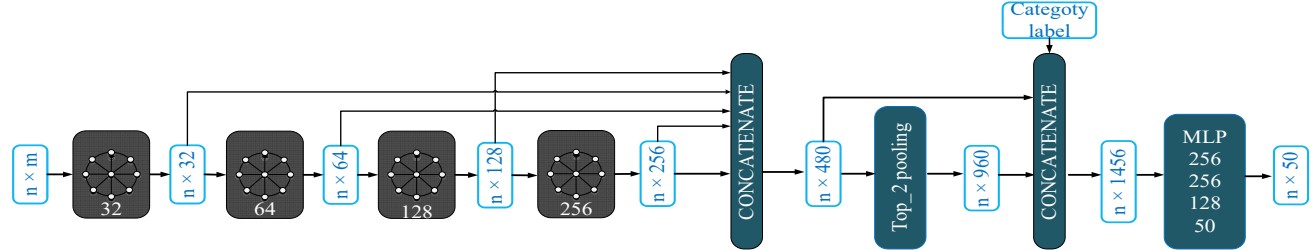

**Figure 10.** The architecture of OECNN in the ShapeNet Parts segmentation [6] task.

**Table 2.** Segmentation results on the ShapeNet Parts dataset [6].

|  | PointNet [11] | PointNet++ [12] | SSCN [26] | SpiderCNN [14] | PointSIFT+ OECNN [18] | OECNN |
|---|---|---|---|---|---|---|
| aero | 83.4 | 82.4 | 81.6 | 83.5 | 83.3 | 83.1 |
| bag | 78.7 | 79.0 | **81.7** | 81.0 | 79.9 | 79.6 |
| cap | 82.5 | 87.7 | 81.9 | 87.2 | 85.8 | **89.6** |
| car | 74.9 | 77.3 | 75.2 | 77.5 | 77.3 | **79.1** |
| chair | 89.6 | 90.8 | 90.2 | 90.7 | 90.2 | **90.8** |
| ear phone | 73.0 | 71.8 | 74.9 | 76.8 | 77.1 | **78.9** |
| guitar | 91.5 | 91.0 | **93.0** | 91.1 | 90.9 | 91.6 |
| knife | 85.9 | 85.9 | 86.1 | 87.3 | 87.2 | **87.5** |
| lamp | 80.8 | 83.7 | **84.7** | 83.3 | 82.7 | 83.7 |
| laptop | 95.3 | 95.3 | 95.6 | 95.8 | 95.6 | **96.0** |
| motor | 65.2 | 71.6 | 66.7 | 70.2 | 71.3 | **73.0** |
| mug | 93.0 | 94.1 | 92. | 93.5 | 94.0 | **95.3** |
| pistol | 81.2 | 81.3 | 81.6 | **82.7** | 82.4 | 81.8 |
| rocket board | 57.9 | 58.7 | 60.6 | 59.7 | 59.6 | **62.7** |
| skate | 72.8 | 76.4 | **82.9** | 75.8 | 75.3 | 76.3 |
| table | 80.6 | 82.6 | 82.1 | 82.8 | 82.2 | **83.0** |
| mIoU | 83.7 | 85.1 | 84.7 | 85.3 | 84.9 | **85.5** |

We show the qualitative results of segmentation on the ShapeNet Part dataset in Figure 11, where ground truth represents the visualization results made by real labels, prediction is the result predicted by the network, and difference represents the misclassified points (red points) between ground truth and prediction. Different colors (ground truth, prediction) represent different part labels. We can see that the segmentation of some points was not very good at the occlusion and the intersection of different parts. This may have lost some effective points for local feature learning.

*4.4. Robustness Test*

In this section, we additionally tested and analyzed the robustness of the OECNN on ModelNet40. We studied the effect of OECNN losing points. Following the settings for the experiments in Section 4.2, we trained a 4-layer OECNN and SpiderCNN with 512, 256, 128, 64, and 32 points as the input data. As shown in Figure 12, as the number of input points decreased, our classification accuracy on ModelNet40 decreased slightly until the number of input points drops to 256. Our classification accuracy was 92.6% when the number of input points was 512. When there were only 32 input points, our OECNN classification accuracy was 87.9%, which was better than that of the SpiderCNN. The disadvantage of our method is that we may not find the corresponding number of points in each direction in a local range, although we use the center point for initialization. This is not conducive to the semantic learning of local features but the comparison shows the effectiveness of our method.

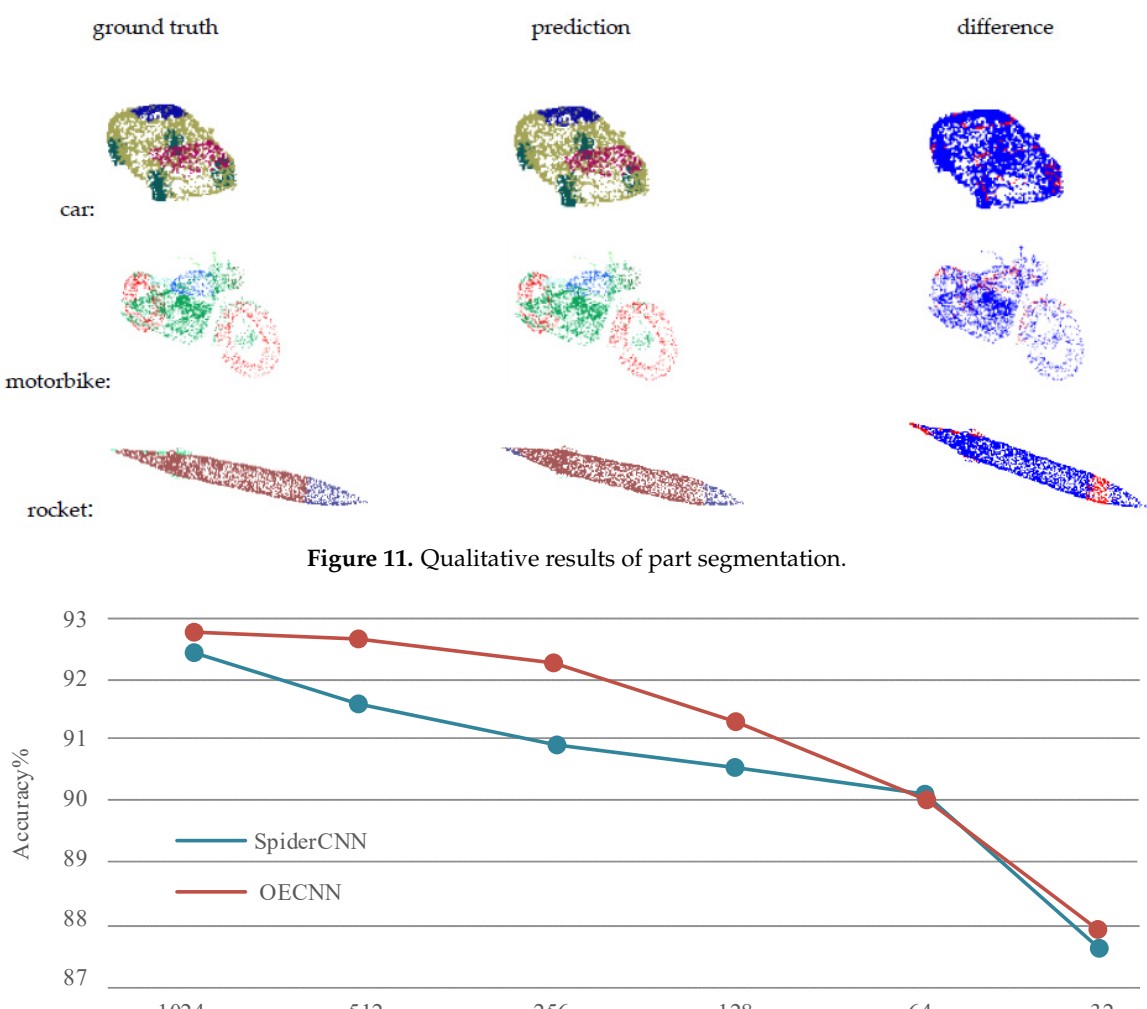

**Figure 11.** Qualitative results of part segmentation.

**Figure 12.** Classification accuracy of OECNN and SpiderCNN with different numbers of points as the input on Model-Net40 [20].

### 4.5. Ablation Experiments

To verify the effectiveness of OEConv, in Table 3, we calculated the results of classification and segmentation when the points in 8 directions were all filled by the center point $p_0$. With these comparisons, we conclude that OEConv, which randomly selects points in 8 directions within a certain range, is key to the performance of OECNN.

**Table 3.** The selection of points in OEConv.

|  | **Classification (Accuracy)** | **Segmentation (mIoU)** |
| --- | --- | --- |
| OEConv (filled by $p_0$) | 91.5% | 84.5% |
| OEConv (random) | 92.7% | 85.5% |

### 4.6. Time and Space Complexity Analysis

Table 4 summarizes space (number of parameters in the network) and time (floating-point operations/sample, forward time) complexity of our classification OECNN. We also compare OECNN to SpiderCNN and PointSIFT (put the convolution operator proposed by PointSIFT into the OECNN) architectures in previous work. While SpiderCNN and PointSIFT achieve high performance, OECNN is more efficient in computational cost (measured by FLOPs/sample and forward time). Besides, OECNN is much more space-

efficient than SpiderCNN in terms of parameters in the network. In the future, we will reduce the amount of network parameters and further study the features effectively.

**Table 4.** Time and space complexity of network architectures for 3D data classification.

| Methods | Parameters | Forward Time | FLOPs/Sample |
|---|---|---|---|
| SpiderCNN [14] | 2.7 M | 0.132 s | 2041 M |
| PointSIFT+OECNN [18] | 1.6 M | 0.116 s | 2109 M |
| OECNN | 1.9 M | 0.052 s | 1425 M |

## 5. Conclusions

In this paper, an orientation-encoding CNN is proposed, which improves the performance of classification and segmentation for unorganized 3D point clouds. First, an orientation-encoding module is used to search for points within a certain range of each point. Subsequently, we convolve the corresponding point set features in several directions to obtain more rich local features of each point. After that, top-k pooling is used to extract the global point set features. OECNN was trained more efficiently with augmented datasets using the proposed scheme. The experimental results show that the proposed method generates a significantly higher classification accuracy (92.7%) on ModelNet40 and achieves an mIoU of 85.5% on the ShapeNet Parts dataset.

**Author Contributions:** Conceptualization, H.L. and W.Z.; methodology, W.Z.; software, W.Z.; resources, X.P.; writing—original draft preparation, W.Z.; writing—review and editing, H.L.; visualization, W.Z.; supervision, X.P.; project administration, H.L. All authors have read and agreed to the published version of the manuscript.

**Funding:** This research was funded by National Key R&D Program of China, grant number 2017YFB0306402, Natural Science Foundation of Hebei Province grant number E2020203188, Key Foundation of Hebei Educational Committee grant number ZD2019039, and Young Talent Program of Colleges in Hebei Province grant number BJ2018018.

**Institutional Review Board Statement:** Not applicable.

**Informed Consent Statement:** Informed consent was obtained from all subjects involved in the study.

**Data Availability Statement:** Not applicable.

**Acknowledgments:** The authors wish to thank the anonymous reviewers for their detailed and constructive comments which are very helpful to the improvement of this paper.

**Conflicts of Interest:** The authors declare no conflict of interest.

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
