# Peer review of "Orientation-Encoding CNN for Point Cloud Classification and Segmentation"

_make, doi:10.3390/make3030031_

Round 1
Reviewer 1 Report
An orientation-encoding convolutional module for point clouds is presented. For each point p_0 one looks for a fixed number of points in eight different directions and inside a sphere around the point p_0. For each direction convolution and top_k pooling is applied. The network is trained for classification and segmentation tasks and these results are compared to different baselines.
It is stated that PointSIFT "needs to traverse all the input points" when looking for the nearest points (section 3). Then you claim that the method is faster because it only searches in radius r for the nearest points. If you do not use a special data structure, doesn't one need to iterate over every point to check whether it is inside the sphere or not? This puts the claimed reduced runtime in question.
There are a few aspect of the proposed method that are not fully clear:
Question 1: In your OE convolutional module you are always looking for a fixed number of points in each direction. What does happen if there are less points in the search direction and inside the sphere of radius r.
Question 2: When you divide the sphere in 8 direction, how are these chosen ? Do you do this with respect to the x-, y- and z-axis? What happens if the angles are different or the data is rotated ?
In the experiment section it is statted that the method improves the results of other models on different datasets. The comparison seems complete and the method improves the results for the classification task as well as the segmentation task. The improvements are for a few cases are relatively low. To underline that the results are better I recommend you to state the accuracy or error of more than one training run (including std.dev.) since you are using the stochastic Adam optimizer, e.g. as [14] did.
Understandability: The grammar and word choice should be improved. Some errors prevent an understanding of ideas and motivation. (i.e: "a kind of rule data" in 2.2, "has been widely studied based on data using neural network iteratively iterated by algorithm" in 2.1, "We suspect that this may be have lost some effective points" in section 4.3)
Citations: You did not fully define top_k pooling in section 3.3 or section 3., where a citation to read more about it is mssing. Also in section 2.1 a citation would be helpful to understand the tasks better.
The SoTa is quite close to ref [14].
Reviewer 2 Report
In this paper, an orientation-encoding (OE) CNN method is proposed for point cloud classification and segmentation. The literature review is extensive for possible readers, and the related work categories is proper. For the method design, the concept of the proposed is clear and can be extended by others in the future. However, for the possible publication, the following points should be enhanced:
- In this paper, the ‘novel’ term is used too many times. The authors should keep the possible readers to think it novel when reading.
- The main concept from the proposed OE convolution to the PointSIFT is clearly drawn in Fig. 1 and Fig. 2, but the descriptions in Sec. 3.1 are still not clear enough. The authors should add block diagrams or pseudo-code in Sec. 3.1 to make the sentences more readable.
- Similarly, the sentences in 3.2 still need additional figures (block diagrams / pseudo-codes) to make it more readable.
- The sentences for describing Max pooling in Sec. 3.3 is ok, when the author mentioned about SpiderCNN, the reference number is missed, the author should add the reference number here, similarly, ModelNet40 in the end of Sec. 3.3 is also missed.
- The results in Fig. 6 and Fig. 8 for the different scales shown that the 0.2 scale achieve the best result, and the authors have some discussion. If the multi-scale architecture cannot provide better results, should the author remove this part for the proposed method?
- In the experimental results, proper citation is needed, for example, in the caption of Fig. 7, the author should add the reference number for ModelNet 40 at the end.
- In Table 1 and Table 2, the authors compared with some of state-of-the-art methods in the years 2017-2019, and most of the methods are published in top conferences like CVPR, ECCV, and NIPS. Is it possible to add one or two more methods for classification and segmentation in the recent 3 years?
- In Fig. 10, the empty part of this figure is large, can the authors use the full words instead of using the abbreviation at the top of the figure? In addition, since the diff is large can the authors add more comparisons (existing methods) and discussions for the reasons?
- The typo in the first page PointnNeg++[12] should be corrected.
Reviewer 3 Report
This paper represents a method for improving the performance of classification and segmentation for unorganized 3D point clouds. The main contribution is that the authors shown the corresponding point set features in directions to obtain more rich local features of each point. The quality of this study is good. But they lost some applications and proof of academic research is unfortunate. This paper presented not clear for some key points. The paper should be revised for resubmission.
- In the summary, the authors should explain the quantitative data for their method. Please consider explain the objects of classification and segmentation. Should explain the advantages or innovations of your method? It does not use the “novel” here.
- There are still some places did not carefully explained. For example: some related applications with their method. For the balanced overview and in order to provide more motivations for the present work, the authors should include some studies of latest examples of three-dimensional convolutional neural network voxelization scenarios, in the introduction section.
Suggested Addition:
In the page 1:
In [6], [7], [8] , [R1] methods, they try to solve this problem by using three-dimensional convolutional neural network voxelization scenarios, but the main challenges of voxel respresentation are spatial sparsity and computational complexity, so in [9], [10], the researchers try to use special methods (such as octree) to solve sparsity problem, but it takes a certain amount of time to convert the point cloud into voxels.
R1: Kai-Chun Li et al.,” Intelligent Identification of MoS2 Nanostructures with Hyperspectral Imaging by 3D-CNN,” Nanomaterials 10(6), 1161 (2020).
- In the Fig.5, the authors should add the real objects to point out architecture of a 4-layer.
- In the section 3.3, please make a figure to point out the difference between the Top_k pooling and Max pooling.
- In the Fig.6, the authors shown three key parameters to improve the performance of the optimized network by using the single variable principle. These results have small difference for each parameters. Has it any important meaning for the comparison?
- In the figure 10, the authors only shown the simple objects. The authors should explain this method can be used for more complicated objects detect or not?
Round 2
Reviewer 1 Report
The (random) point selection process can be explained better, it is not quite clear how that pursues, and how the mentioned rotation plays a role. With the randomness in the overall algorithm I think a std-dev-study is needed and not just state the best performances.
Author Response
Thank you very much for your suggestion. First of all, the key factor of the point selection process is the division of the spherical area, and the selected random points are relevant according to the data processing stage. Secondly, the rotation operation is also to improve the overall performance of the network and reduce the occurrence of accidents. Finally, although the performance of our network is not the best, it has been greatly improved on the basis of other related papers.
Reviewer 2 Report
Most of my raised points are well-addressed.
This paper is ready to be published.
Author Response
Thank you very much for your approval, we will continue to study in depth in this direction.
Reviewer 3 Report
The authors reply all my comments. This article can be accepted by MAKE.
Author Response

(The authors gave the same response as above.)
